# Age-Dependent Growth-Related QTL Variations in Pacific Abalone, *Haliotis discus hannai*

**DOI:** 10.3390/ijms241713388

**Published:** 2023-08-29

**Authors:** Kang Hee Kho, Zahid Parvez Sukhan, Shaharior Hossen, Yusin Cho, Won-Kyo Lee, Ill-Sup Nou

**Affiliations:** 1Department of Fisheries Science, Chonnam National University, Yeosu 59626, Republic of Korea; sukhan1026@jnu.ac.kr (Z.P.S.); shaharior@jnu.ac.kr (S.H.); 218179@jnu.ac.kr (Y.C.); wklee196@jnu.ac.kr (W.-K.L.); 2Department of Horticulture, Sunchon National University, Suncheon 57922, Republic of Korea; nis@sunchon.ac.kr

**Keywords:** QTL, SNP, QTL variation, growth-trait, age, Pacific abalone, *Haliotis discus hannai*

## Abstract

Pacific abalone is a high-value, commercially important marine invertebrate. It shows low growth as well as individual and yearly growth variation in aquaculture. Marker-assisted selection breeding could potentially resolve the problem of low and variable growth and increase genetic gain. Expression of quantitative trait loci (QTLs) for growth-related traits, viz., body weight, shell length, and shell width were analyzed at the first, second, and third year of age using an F1 cross population. A total of 37 chromosome-wide QTLs were identified in linkage groups 01, 02, 03, 04, 06, 07, 08, 10, 11, 12, and 13 at different ages. None of the QTLs detected at any one age were expressed in all three age groups. This result suggests that growth-related traits at different ages are influenced by different QTLs in each year. However, multiple-trait QTLs (where one QTL affects all three traits) were detected each year that are also age-specific. Eleven multiple-trait QTLs were detected at different ages: two QTLs in the first year; two QTLs in the second year; and seven QTLs in the third year. As abalone hatcheries use three-year-old abalone for breeding, QTL-linked markers that were detected at the third year of age could potentially be used in marker-assisted selection breeding programs.

## 1. Introduction

Abalone is a univalve marine molluscan gastropod from the *Haliotis* genus. It has a worldwide distribution in tropical and temperate zones including South Korea [1]. Pacific abalone (*Haliotis discus hannai*) is the most popular marine food resource among the abalone species in South Korea and Southeast Asian countries. The bioactive molecules in the muscle have potential antioxidant and anti-cancerous activities, which make abalone a popular seafood on national and international markets [2,3]. Pacific abalone possesses a large edible adductor muscle that turns it into an attractive aquaculture species. Although land-based abalone farming began in the 1980s in Korea, intensive sea-cage aquaculture production massively expanded in the 2000s [4]. However, the slow and variable growth rate of abalone is a major problem for abalone aquaculture [5].

Growth is an economically important trait in the invertebrate aquaculture industry as it is directly related to production. Abalone aquaculture is facing a critical problem in terms of low growth rate. Further, it is also evident that abalone shows variable growth, i.e., the growth of abalone individuals in a stock varies significantly [5,6]. Therefore, an improvement in the growth rate and homogenous growth of all individuals in a stock is desirable in the abalone farming industry. At present, the entire abalone aquaculture industry uses hatchery-produced abalone seed. Abalone hatcheries produce seeds using conventional selection breeding [7]. Conventional phenotypical selection breeding substantially improves the growth rate of abalone. However, the use of molecular markers and marker-assisted selection (MAS) breeding could resolve the problem of low and variable growth and increase genetic gain [8]. Molecular and marker-assisted breeding is successfully used in many aquaculture species including greenshell mussel [9], Pacific oyster [10], noble scallop [11], whiteleg shrimp [12], salmon [13], Japanese flounder [14], and rainbow trout [15] to achieve higher growth and disease resistance. Regarding abalone, several studies performed genetic analysis and concluded that growth-related traits can be improved using MAS breeding [7,16]. However, the development of potential markers for growth-related traits across ages in abalone has not yet been studied.

It is well-known that economically important quantitative traits of aquatic animals such as body weight, body length, body size, growth rate, etc., are extremely variable characteristics. These traits are influenced by environmental factors, food, and multiple growth-related genes [17]. Changes in the genetic control of growth are common features in aquatic animals including invertebrates. These changes generally occur with the progression of age in a stock and among different families. For example, it is evident that fluctuations in growth occurring at different stages of the life cycle in the guppy are influenced by different genetic controls [18]. Growth-related traits are mainly controlled by quantitative trait loci (QTLs). QTLs are genetic regions linked to a single gene or gene clusters that influence phenotypic variation in a complex trait, often through genetic interactions with each other and the environment [19,20]. Markers linked to QTLs are used in MAS and can increase the accuracy of selection and achieve genetic improvement with direct and prompt selection [21,22]. However, it has been observed that QTL expressions are not stable with the progression of age in animals and plants [23,24,25]. Along this line, progenies displaying fast and/or higher growth in the early stages of life may not show a similar growth rate in successive years. These variations in age-related growth and QTL expression may occur due to variations in the genetic control of growth as age increases [25]. Therefore, it is important to consider the influences of the genetic control of growth in abalone at different ages in the same population in breeding and aquaculture.

Growth-related QTL studies have been performed on several commercially important molluscan species, including the Pacific abalone [7,16], small abalone [26], blacklip abalone [27], Manila clam [28], Pacific oyster [29], and bay scallop [30]. On the other hand, QTL variations across ages and families have been reported in several mammals and fishes, including the gilthead seabream [23], blackface sheep [24], nine-spined stickleback [31], chicken [32], and common carp [33]. However, no report on QTL variations across ages in invertebrate species has been published. Two studies on growth-related QTLs in the Pacific abalone have been reported so far, which showed different QTL expressions [7,16]. Our group previously reported two growth-related QTLs in one-year-old Pacific abalone [7]. Herein, we aimed to observe the variations in growth-related QTLs across ages in an F1 population of Pacific abalone.

## 2. Results

### 2.1. Variations in Phenotypic Traits

The mean phenotypic values for body weight (BW), shell length (SL), and shell width (SW) of the F1 population at different ages that were used for the QTL analysis are presented in Table 1. A total of 94 F1 progenies were used in the QTL analysis at each age. The phenotypic values for each F1 progeny are present in Appendix A. The mean value for all growth-related traits significantly increased with increasing age. Changes in the BW of a few randomly selected F1 individuals at different ages are illustrated in Figure 1. The graph explained that some individuals showing lower weight in the first year reached a higher weight gain in the third year, and vice versa.

Correlation coefficients among all growth-related traits in the F1 population at different ages are presented in Table 2. The results showed a positive correlation among all growth-related traits. The correlations among the first, second, and third years continuously decreased for each trait. Additionally, for each trait, higher correlations were observed between the first and second year: 0.738 for BW, 0.726 for SL, and 0.631 for SW, which were strongly correlated. However, the highest correlation was observed between SW and SL in the first year (0.964). Year-wise correlations among the growth-related traits are presented in Appendix A.

### 2.2. Genetic Linkage Map

Three integrated genetic linkage maps were constructed over three years using 3314, 2971, and 2279 effective single nucleotide polymorphism (SNP) markers, respectively (Table 3; Appendix A). The linkage maps were constructed using the SNP markers and phenotypic data on the F1 population of each year. Afterward, a total of 18 linkage groups (LGs) were generated. The consensus linkage maps expressed a map distance of 1747.023 cM, 1648.91 cM, and 1460.986 cM for the three years, respectively. However, the average distances between adjacent markers were 0.55 cM, 0.63 cM, and 0.75 cM for the three years, respectively (Table 3). Details of the SNP markers at the three different ages are presented in Appendix A, respectively.

### 2.3. QTL Analysis

A summary of the QTL locations in the linkage maps at the different ages is displayed in Figure 2. A total of two QTLs were detected at the first year, eleven QTLs were detected at the second year and twenty-four QTLs were detected at the third year for growth-related traits. 

#### 2.3.1. QTLs Associated with Body Weight (BW) at Different Ages

The QTLs for BW at different ages are presented in Table 4 and Figure 2. A total of 29 chromosome-wide QTLs were identified on LG01, LG02, LG03, LG04, LG06, LG07, LG08, LG10, LG11, LG12, and LG13. One QTL for BW was detected on LG03 (in the third year), LG04 (in the third year), LG06 (in the third year), and LG08 (in the third year) each; two QTLs on LG10 (in the first year) and LG11 (in the third year) each; three QTLs on LG01 (in the second year) and LG07 (in the second year) each; four QTLs on LG02 (in the second year) and LG12 (in the third year) each; and seven QTLs on LG13 (in the third year). A higher number of BW QTLs were detected for the mature stage of growth (seventeen QTLs in the third year) than for the early growth stage (two QTLs in the first year). The average polymorphism information content (PIC, which relates to the average allelic diversity in all markers) for the markers reached 0.311 in the first year and 0.375 in the second and third years. The PIC values ranged from 0.305 to 0.311 in the first year, 0.276 to 0.375 in the second year, and 0.225 to 0.375 in the third year. The logarithm of odds (LOD) values ranged from 25.962 to 29.006 in the first year, 3.005 to 3.823 in the second year, and 2.000 to 3.415 in the third year of age (Table 4 and Figure 3).

#### 2.3.2. QTLs Associated with Shell Length (SL) at Different Ages

A total of 20 QTLs for the SL at different ages were detected on LG02, LG04, LG06, LG07, LG08, LG10, LG11, and LG12 (Table 5 and Figure 2). For the first year of age, two QTLs for SL were detected on LG10. For the second year of age, two QTLs on LG02, one QTL on LG06, and two QTLs on LG07 were detected. On the other hand, for the third year of age, ten QTLs were detected in six different LGs: one QTL on LG04, two QTLs on LG06, one QTL on LG08, two QTLs on LG10, three QTLs on LG11, and four QTLs on LG12. As observed for BW QTLs, a higher number of SL QTLs were also detected for the mature age (ten QTLs in the 3rd year) than for the early age (two QTLs in the first year). The average PIC for markers related to SL reached 0.311 in the first year, 0.322 in the second year, and 0.375 in the third year. The PIC values ranged from 0.305 to 0.311 in the first year, 0.276 to 0.325 in the second year, and 0.201 to 0.375 in the third year. The LOD values ranged from 25.962 to 29.006 in the first year, 3.005 to 3.823 in the second year, and 2.000 to 3.415 in the third year of age (Table 5 and Figure 3).

#### 2.3.3. QTLs Associated with Shell Width (SW) at Different Ages

The QTL results for SW at different ages are presented in Table 6 and Figure 2. A total of 17 QTLs were detected on LG02, LG04, LG06, LG10, LG11, and LG12. Two QTLs on LG10 and LG02 were detected in the first and second years of age, respectively. However, thirteen QTLs were detected in the third year of age in six different LGs: two QTLs on LG02, one QTL on LG04 and LG06 each, two QTLs on LG10, three QTLs on LG11, and four QTLs on LG12. Like BW and SL, for SW QTLs, a lower number of QTLs was detected in the early stage of age (two QTLs in the first year) and a higher number of QTLs was detected at the mature stage of age (thirteen QTLs in the third year). The average PIC for the markers related to SW reached 0.311 in the first year, 0.385 in the second year, and 0.375 in the third year. The PIC values ranged from 0.305 to 0.311 in the first year, 0.276 to 0.311 in the second year, and 0.201 to 0.375 in the third year. The LOD values ranged from 30.886 to 33.883 in the first year, 4.741 to 4.763 in the second year, and 2.544 to 3.817 in the third year of age (Table 6 and Figure 3).

## 3. Discussion

The present study reported variations in QTL expression at different ages in Pacific abalone. This is the first report addressing QTL analyses at different ages for growth-related traits in abalone and other invertebrate species. The QTL analyses were performed using SNP markers that were generated from genotype-by-sequencing (GBS) data at three developmental time points (first, second, and third year of age) in a full-sib family of Pacific abalone. Finally, several independent age-dependent QTLs and linked SNP markers were detected at different ages.

The present results demonstrate that the QTL expression for growth-related traits, such as BW, SL, and SW, were different across ages in Pacific abalone. Variations in QTL expression in growth-related traits at different ages have been reported in Scottish blackface sheep [24], the radiata pine [25], the nine-spined stickleback [31], chicken [32], mice [34], and the rubber tree [35]. None of the detected QTLs were expressed in all ages of Pacific abalone, which means there is no common QTL that affects growth traits at every age. A similar QTL variation in growth-related traits across ages was reported in fish and plants including the nine-spined stickleback [31] and radiata pine [25]. Further, the number of QTLs detected for each trait varied with age. At the early stage of life (first year), a lower number of QTLs was detected compared with the mature stage of life (second year) in Pacific abalone. Growth variation in the QTL progeny of abalone across ages was also observed. This growth variation may have resulted from the activation and repression of genes responsible for changes in growth over time. The timing of ontogenetic cycles of gene activity determines the total length of the developmental process as well as the timing of critical steps during various ontogenetic processes [36]. Therefore, the changes in QTL expression across the ages may be due to growth variations across the ages, which are governed by the activation and repression of associated genes. Further, the results might suggest that genes responsible for growth at different ages may be different and act independently.

Growth is a highly polygenic trait; thus, the detection of QTLs on several chromosomes is expected. Phenotypic variation occurs in a population with the effect of polygenic control, which is largely determined by genetic variation at multiple QTLs. In addition, the inherent genetic architecture of quantitative traits may differ across development and in different environments [17,37]. It is difficult to analyze QTL mapping for polygenic traits that include low statistical power to detect significant QTLs [38]. In the present study, the effect size and amount of PVE by age-specific QTLs in abalone were relatively low. This result suggests that BW, SL, and SW are largely polygenic quantitative traits in Pacific abalone. Therefore, the detection power for QTL mapping analysis in the present study could be limited by sample size and statistical power to detect enough causative loci at each age. Variation in QTL expression at different ages, as observed in the present study, may also occur due to the polygenic control of growth traits. A similar result also explained this for nine-spined stickleback [31].

In an aquaculture system, abalone grows in two culture environments. First, abalone larvae are cultured in a land-based rearing tank for 10 to 12 months. Later, they are transferred to a sea-cage grow-out system and cultured for another 2–3 years. The change in the culture environment may be another reason for QTL variations at different ages in Pacific abalone. It is apparent that the genetic architecture of quantitative traits may vary in different environments [17]. In juvenile Atlantic salmon, QTL variations occurred among hatchery populations and wild populations [39]. In another study, it was reported that QTL variations transpired in freshwater and marine populations of nine-spined stickleback [40].

In the present study, variations in multiple-trait QTL locations for three growth-related traits were detected each year. The term ‘multiple-trait QTL’ refers to the detected QTL that affected all three traits analyzed in this study. Two multiple-trait QTLs were detected in the first year in LG10; two multiple-trait QTLs were observed in the second year in LG02; and one QTL was noticed in LG04, two QTLs in LG11, and four QTLs in LG12 in the third year of age, which were detected as multiple-trait QTLs for each growth-related trait (Figure 2). The multiple-trait QTLs in three growth-related traits suggest that these growth traits might be controlled by a common chromosomal region. However, the changes in the location of multiple-trait QTLs for growth traits in each year suggest that the growth traits for different ages might be controlled by a different common chromosomal region at different ages. Further, the variation in QTL expression across ages suggests that the genetic control of growth traits might be controlled by a different genomic region at different ages. Multiple-trait QTLs across different growth traits have also been reported for Pacific abalone [7], small abalone [26] and Asian seabass [41]. In addition to the multiple-trait QTLs, the QTL for each single trait was also detected in different LGs, suggesting that different growth traits are also controlled by several chromosomal regions. However, the results clearly indicate that some QTLs have more general effects on all growth-related traits for different ages, whereas other QTLs have specific effects on specific traits. Further, multiple-trait QTLs for growth traits were detected in the telomeric region in the first and third years of age. The recombination rate in the chromosome is believed to be higher in the telomeric region compared with the centromeric region of the chromosome [42,43], and a higher residual heterozygosity is expected for the telomeric region [44]. Thus, the presence of QTLs in the telomeric region is considered a robust result and suggests a more advanced segregation and recombination rate.

MAS breeding is one of the most important applications that can benefit from QTL mapping. Traditional selection breeding techniques use information from a known pedigree and phenotypes for a selected population, which only allows phenotypic improvement. However, genetic improvement can only be gained by using MAS. MAS could potentially increase the genetic gains by 8–38% over traditional selection breeding [45]. In MAS, QTL-linked markers associated with traits of interest allow breeders to accurately select breeding individuals based on genotype. In the present study, all detected QTLs and markers were age-specific. Therefore, it is important to select age-specific QTL-linked markers in MAS breeding programs of abalone. Generally, 3-year-old abalone is used in breeding programs. Therefore, it is recommended that MAS breeding using 3-year-old abalone should use QTL-linked markers obtained from the third year of age.

## 4. Materials and Methods

### 4.1. Collection of Parents and Production of Mapping Family

#### 4.1.1. Collection of Parent Abalone

To produce a mapping family, male and female parents of Pacific abalone of the same age were collected from two different geographically distant areas in the Republic of Korea. Male and female parents were collected from the Boryeong (36°18′56.3″ N 126°26′17.8″ E) and Wando (34°19′12.8″ N 126°39′02.6″ E) regions of Korea, respectively. About 100 male and female parent abalone were collected and nurtured in an abalone hatchery in Dolsan-eup, Yeosu, Republic of Korea. Abalones were nurtured in a rearing tank with running seawater and continuous aeration and fed with seaweed (*Laminaria japonica*) to satiation. Parent abalones were nurtured until reaching a gonadally mature condition. 

#### 4.1.2. Production of Mapping Family

The F1 family was produced in May 2019 using the collected parent abalones, which were used as a mapping family to construct linkage maps at different ages. Among the parents, the female was small (94.97 mm in length and 95.64 g in weight) and the male was large (105.21 mm in length and 115.92 g in weight) in size. At the fully mature gonadal stage, induced spawning of the parent abalones was performed using the UV-irradiation method [46,47]. Briefly, reproductively mature parents were first induced under sunlight for 60 min in an upside-down direction and then 30 min in the downside-up direction. Heat-induced parent abalones were then placed in a 20 L spawning bucket with UV-irradiated seawater. Spawning buckets were placed in the dark with continuous aeration. After gametes were released, fertilization was performed, and the larvae were spread in a larval-rearing tank having a pre-grown diatom plate. Around 3 months later, 15,000 F1 abalone progenies were transferred to a separate rearing tank and grown until 12 months of age. A flow diagram showing the timeline and production of mapping family, rearing of F1 progenies, and QTL analysis is presented in Figure 4.

### 4.2. Husbandry of Abalone F1 Population

#### 4.2.1. Husbandry in the Larval Rearing Tank

Abalone juveniles were reared in a tank with continuous aeration and running seawater with a flow rate of 1.5 L/min. Water quality parameters were monitored every day. Abalone juveniles were fed daily to satiation with commercial abalone juvenile feed (Gold Ocean Premium Abalone Feed, JOEUN Corp., Wando, Republic of Korea). The rearing tanks were cleaned every alternative day by changing all water and flushing water.

#### 4.2.2. Husbandry in Sea Cage

After 12 months of rearing, abalone juveniles were transferred to a sea cage in Jindo-gun, South Korea. Abalone juveniles were fed with live seaweed (*Laminaria japonica*) to satiation. Abalones were reared in a sea cage until three years of age. All abalone were tagged by using a sealing tag. 

### 4.3. Sample Collection and Trait Measurement

Samples were collected at three time points: the first, second, and third year of age from the F1 family. Samples at the first year age were collected during the transfer of abalone from the tank condition to the sea cage. Samples at the second and third years of age were collected during the completion of the second and third years of age. In total, 94 F1 samples (47 largest and 47 smallest) each year were randomly selected, as described previously [7,27,48]. Three quantitative traits related to the growth of abalone, including body weight (BW), shell length (SL), and shell width (SW), were measured for each abalone sample with a digital weighing machine (SPX223KR; OHAUS, Parsippany, NJ, USA) and a digital caliper (CD-P20S, Mitutoyo Corp., Kanagawa, Japan) to the nearest g and mm, respectively. All raw data for trait measurements at different ages are presented in Appendix A. Tissue samples were collected from a regenerative organ, the cephalic tentacle (CT), of each abalone. An incision was made in a portion of the CT from each abalone, and then the CT was washed with 1× phosphate-buffered saline (PBS) and immediately snap-frozen in liquid nitrogen. CT samples were stored at –80 °C until extraction of genomic DNA (gDNA). After sample collection, all abalone were returned to the grow-out cage. CT samples of parent abalone were also collected previously during induced spawning.

### 4.4. Genotyping-by-Sequencing (GBS) Library Preparation, Sequencing, and SNP Genotyping

#### 4.4.1. Extraction of Genomic DNA (gDNA)

The gDNA of all CT samples from each F1 abalone in the first, second, and third years of age and the parent abalone were extracted using the CTAB (Cetyl trimethylammonium bromide) method as described previously [7,49]. Briefly, 200 µL of CTAB extraction buffer (0.1 m Tris-HCl pH 8.0, 20 mm EDTA pH 8.0, 1.4 M NaCl, 2% CTAB w/v) containing 73 µL 2-mercaptoethanol was added to 100 mg of the cryogenically grounded fine power of each CT sample and vortexed. The suspension was then incubated in a water bath at 65 °C for 60 min. The incubated sample was then centrifuged for 5 min at 14,000× *g*. The supernatant was then transferred to a new tube and incubated at 37 °C for 20 min after adding 5 µL of RNase A. Later, an equal volume of phenol/chloroform/isoamyl alcohol (25:24:1) was added to the mixture, vortexed for 5 s, and incubated for 5 min at room temperature. The mixture was then centrifuged for 5 min at 13,000× *g*. The aqueous phase supernatant was then transferred to a new tube, and 200 µL of phenol/isoamyl alcohol (24:1) was added to the supernatant, vortexed for 5 s, incubated for 5 min at room temperature, and then centrifuged for 5 min at 13,000× *g*. The clear aqueous supernatant was transferred to a new tube and 2.5 volumes of ice-cold absolute ethanol were added, mixed well, and incubated for 30 min at −70 °C. To obtain a DNA pellet, samples were centrifuged at −4 °C for 30 min. The supernatant was then removed, and the pellet was washed with ice-cold 70% ethanol. After washing, the ethanol was discarded, and the residual ethanol residue was removed using air-drying. Finally, DNA was obtained by dissolving the pellet in 20 µL of Tris-EDTA buffer. The concentration of DNA samples was measured using a NanoPhotometer spectrophotometer (Implen, Westlake Village, CA, USA). Finally, the integrity of the DNA samples was evaluated using 1.2% agarose gel electrophoresis.

#### 4.4.2. GBS Library Preparation and Sequencing

The GBS library was prepared for each year’s DNA samples. The GBS library was prepared using a double digestion protocol with two restriction enzymes (PstI and MspI), as described previously [7,50]. The quality of the GBS libraries was checked, and agarose gel electrophoresis was performed. The pooled GBS libraries were then sequenced using the paired-end read method using an Illumina HiSeq X sequencing platform (Illumina Inc., San Diego, CA, USA). Next-generation sequencing was performed at SEEDERS, Daejeon, Republic of Korea. A summary of the GBS raw sequence data is presented in Appendix A.

#### 4.4.3. Cleaning of Sequences

Raw sequences obtained from HiSeq X sequencing were demultiplexed into 96 samples according to barcode sequences. Barcode and adapter sequences were removed using Cutadapt (version 1.8.3) software [51]. Low-quality sequences were trimmed using DynamicTrim and LengthSort programs of SolexaQA (v.1.13) package [52]. An overview of GBS sequence data after cleaning is presented in Appendix A.

#### 4.4.4. Detection of Raw SNPs, SNP Filtering, and Genotyping

Cleaned sequences were aligned to the reference genome sequence of Pacific abalone, which contained 80,032 scaffolds (Appendix A). Sequence alignment was performed using the Burrows–Wheeler Aligner (BWA) program version 0.6.1-r104 [53]. Raw SNPs were detected, and consensus sequences were extracted using SAMTools ver. 0.1.16 [54]. Later, the detected raw SNPs were validated using an in-house script of SEEDERS [55]. The validated SNPs were then classified as homozygous (SNP read rate ≥ 90%), heterozygous (40% ≤ SNP read rate ≤ 60%), or ambiguous (20% ≤ SNP read rate ≤ 40% or 60% < SNP read rate < 90%). The SNP markers between two parents were classified into three segregation patterns including <nn×np>, <lm×ll>, and <hk×hk>. The markers showing significantly distorted segregation (*p* < 0.05) were removed based on the detection of abnormal bases. The markers were grouped with a minimum logarithm of odds (LOD) score of 6.0 and a recombination frequency of 0.45. A summary of SNP marker-related data obtained from the sequence analysis for the different ages is presented in Appendix A.

### 4.5. Construction of Genetic Linkage Map and QTL Analysis

SNP markers that satisfied the expected Mendelian segregation ratio were used for mapping. The ratios of marker segregation were calculated using the Chi-square test. A regression mapping algorithm was used to build the linkage map. Linkage analysis and genetic linkage map construction were performed using JoinMap 4.1 [56] with population type cross-pollination (CP) [57] under the condition of a LOD of 6.0 or higher with a maximum distance of 30 cM. A regression mapping algorithm was used to build the linkage map. Map distances were calculated in centiMorgans (cM) according to the Kosambi mapping function [58].

### 4.6. QTL Mapping

QTL analysis for growth-related traits was performed using R/qtl [59]. First, to identify potential QTLs, interval mapping (IM) was performed. Then, multiple QTLs were identified using a multiple QTL model (MQM) with the stepwiseqtl function. Finally, significant QTLs were identified with a significant LOD threshold of 3 at a 95% confidence level. Significant thresholds were determined using 1000 permutations (*p* < 0.05). The results from the QTL analysis were used to construct a QTL map, and their positions were used in a default model. Markers located at the peak LOD value of the QTL were identified as QTL-linked markers. A QTL analysis using the 1st-year data was previously published [7]. A few data from the 1st year analysis were also included in this manuscript to examine the variations in QTL expressions in different years. 

## 5. Conclusions

The purpose of this study was to identify QTLs and linked SNP markers at different ages that are associated with growth-related traits in Pacific abalone so that MAS could be incorporated into the selective breeding program of abalone. Overall, many QTLs for growth-related traits were detected at different ages. None of the QTLs detected at any one age were expressed at all three age groups, which means all detected QTLs are age-specific. The detected QTLs have single-trait and/or multiple-trait effects at different ages. Age-specific growth-related QTL variations may be associated with specific genes, and the activation and repression of these genes coordinate the developmental process at different ages. The present result of the QTL analysis reported several growth-related QTL-linked SNP markers at different ages that will open a potential window in MAS breeding programs in Pacific abalone.

## Figures and Tables

**Figure 1 ijms-24-13388-f001:**
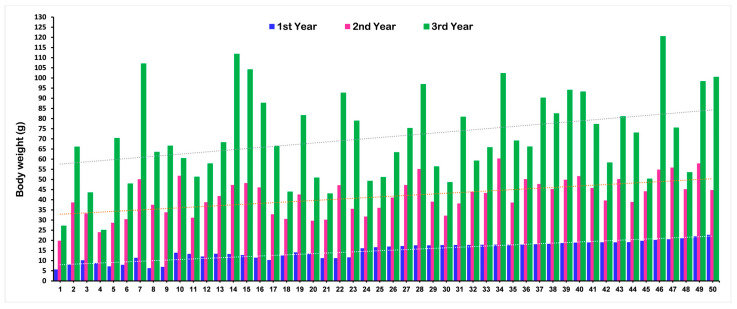
Changes in growth (body weight) for the F1 population of Pacific abalone at different ages.

**Figure 2 ijms-24-13388-f002:**
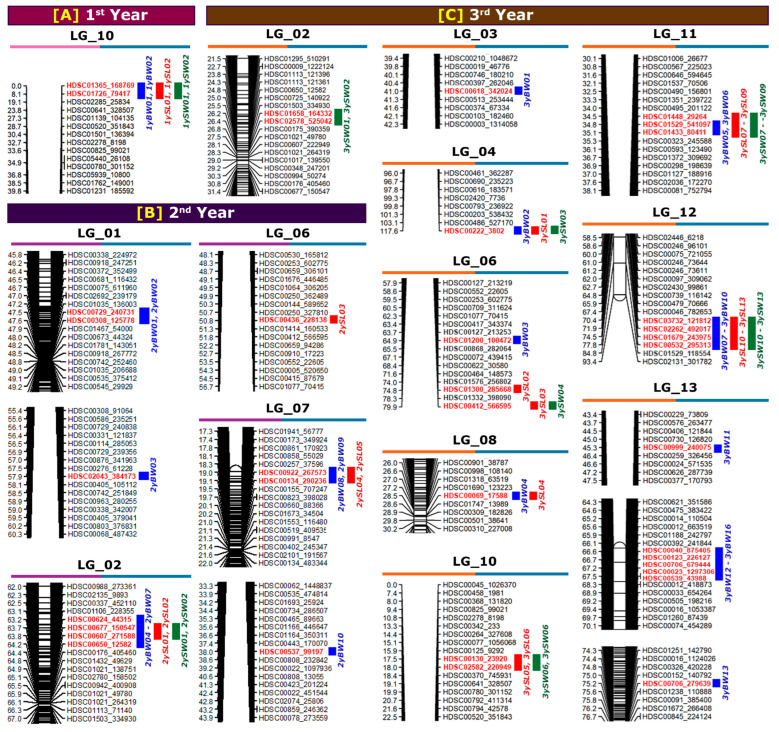
Localization of detected quantitative trait loci (QTLs) in linkage groups (LGs) at different ages of Pacific abalone. (**A**) First year of age, (**B**) second year of age, and (**C**) third year of age. Blue, red, and green colored letters and bars represent QTLs for body weight, shell length, and shell width, respectively. The photograph only shows the selected region of LGs where the QTL was detected. Raw linkage map figures are presented in Appendix A. The number on the left side and the right side of the LGs denotes the marker position (cM, centimorgan) and marker number, respectively. The first-year data (**A**) were adapted from Kho et al., 2021 [7].

**Figure 3 ijms-24-13388-f003:**
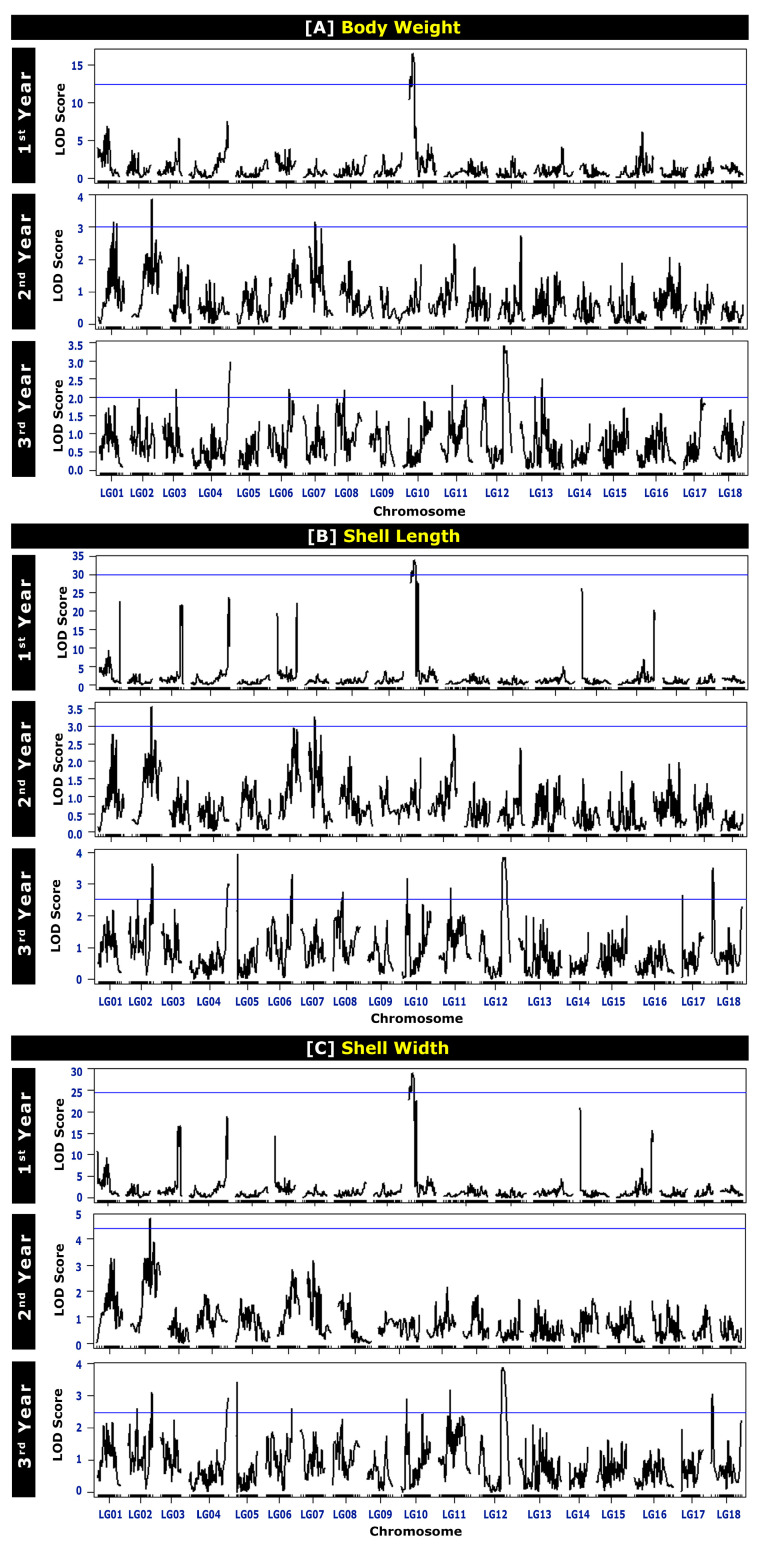
Logarithm of odds (LOD) profile of growth-related traits: (**A**) body weight, (**B**) shell length, and (**C**) shell width of Pacific abalone at different ages. The first-year data were adapted from Kho et al., 2021 [7].

**Figure 4 ijms-24-13388-f004:**
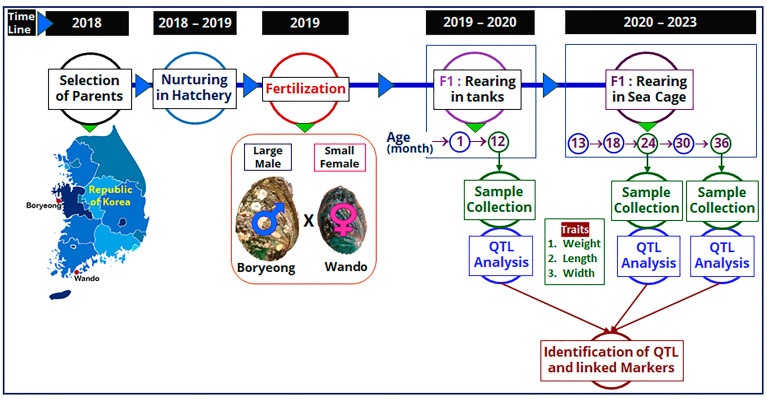
Flow diagram showing the timeline and production of mapping family, rearing of F1 progenies, and QTL analysis for detection of QTLs in Pacific abalone at different ages.

**Table 1 ijms-24-13388-t001:** Phenotypic values of growth-related traits at different ages for the F1 population of Pacific abalone.

Age	Body Weight (g)		Total Length (mm)		Total Width (mm)	
	Mean	SD *	Mean	SD	Mean	SD
First Year	15.28	6.82	48.95	9.42	33.13	6.13
Second Year	33.86	12.96	67.03	10.00	44.96	7.72
Third Year	72.03	21.50	83.31	10.03	55.58	6.97

* SD: standard deviation.

**Table 2 ijms-24-13388-t002:** Pearson’s correlation coefficients for all growth-related traits at different ages.

Traits	BW1	BW2	BW3	SL1	SL2	SL3	SW1	SW2	SW3
BW1	1								
BW2	0.738 **	1							
BW3	0.290	0.156	1						
SL1	0.961 **	0.745 **	0.157	1					
SL2	0.695 **	0.937 **	0.074	0.726 **	1				
SL3	0.437 **	0.469 **	0.260 *	0.508 **	0.434 **	1			
SW1	0.949 **	0.702 **	0.135	0.964 **	0.680 **	0.463 **	1		
SW2	0.625 **	0.776 **	0.010	0.672 **	0.753 **	0.482 **	0.631 **	1	
SW3	0.313 **	0.375 **	0.195	0.371 **	0.334 **	0.669 **	0.337 **	0.633 **	1

Trait abbreviations: BW1, BW2, BW3, body weight (g); SL1, SL2, SL3, shell length (mm); SW1, SW2, SW3, shell width (mm) at the first, second, and third year, respectively. **, correlation is significant at the 0.01 level (2-tailed); *, correlation is significant at the 0.05 level (2-tailed).

**Table 3 ijms-24-13388-t003:** Summary of a linkage map for Pacific abalone at different ages.

Linkage Group	First Year *			Second Year			Third Year		
	No. of Mapped Markers	Length (cM)	Ave. Marker Interval (cM)	No. of Mapped Markers	Length (cM)	Ave. Marker Interval (cM)	No. of Mapped Markers	Length (cM)	Ave. Marker Interval (cM)
LG01	206	71.768	0.35	161	82.487	0.51	113	68.163	0.60
LG02	152	81.334	0.54	134	95.684	0.71	92	56.473	0.61
LG03	222	80.119	0.36	237	67.995	0.29	194	58.785	0.30
LG04	212	131.385	0.62	167	97.662	0.58	190	101.311	0.53
LG05	167	108.524	0.65	184	111.115	0.60	136	62.984	0.46
LG06	160	68.593	0.43	176	72.623	0.41	124	79.907	0.64
LG07	148	79.892	0.54	151	74.229	0.49	103	68.240	0.66
LG08	142	107.082	0.75	123	104.333	0.85	97	79.879	0.82
LG09	112	95.413	0.85	76	131.223	1.73	54	76.833	1.42
LG10	233	92.076	0.40	239	92.379	0.39	183	87.043	0.48
LG11	225	148.86	0.66	189	80.865	0.43	153	77.477	0.51
LG12	180	104.492	0.58	98	85.009	0.87	59	93.447	1.58
LG13	297	130.62	0.44	264	106.487	0.40	243	130.929	0.54
LG14	196	97.554	0.50	204	87.166	0.43	91	55.392	0.61
LG15	191	123.956	0.65	210	122.722	0.58	159	90.525	0.57
LG16	176	89.218	0.51	186	104.947	0.56	157	115.88	0.74
LG17	197	63.172	0.32	99	61.642	0.62	73	64.643	0.89
LG18	98	72.965	0.74	73	70.342	0.96	58	93.075	1.60
Total	3314	1747.023	0.55	2971	1648.910	0.63	2279	1460.986	0.75

* The first-year data were adapted from Kho et al., 2021 [7].

**Table 4 ijms-24-13388-t004:** Quantitative trait loci (QTLs) for body weight (BW) in a full-sib family of Pacific abalone at different ages.

Age	QTL	Linkage Group	Nearest Marker	Position (cM)	LOD *^1^	PVE *^2^	He *^3^	PIC *^4^	No. of Genes *^5^
*First Year* *^6^									
	*1yBW01*	LG10	HDSC01365_168769	0.000	25.962	0.499	0.375	0.305	3
	*1yBW02*	LG10	HDSC01726_79417	8.104	29.006	0.570	0.386	0.311	1
*Second Year*									
	*2yBW01*	LG01	HDSC00729_240731	47.456	3.005	0.137	0.337	0.280	8
	*2yBW02*	LG01	HDSC00308_125778	47.598	3.201	0.145	0.500	0.375	4
	*2yBW03*	LG01	HDSC02043_384173	57.900	3.144	0.143	0.380	0.308	7
	*2yBW04*	LG02	HDSC00624_44315	63.182	3.156	0.143	0.500	0.375	4
	*2yBW05*	LG02	HDSC00677_150547	63.714	3.806	0.170	0.385	0.311	1
	*2yBW06*	LG02	HDSC00607_271588	63.759	3.823	0.171	0.331	0.276	5
	*2yBW07*	LG02	HDSC00650_12582	64.208	3.070	0.140	0.370	0.301	1
	*2yBW08*	LG07	HDSC00922_267573	19.045	3.272	0.148	0.404	0.322	1
	*2yBW09*	LG07	HDSC00134_290236	19.118	3.220	0.146	0.395	0.317	2
	*2yBW10*	LG07	HDSC00537_99197	38.002	3.030	0.138	0.433	0.339	3
*Third Year*									
	*3yBW01*	LG03	HDSC00618_342024	41.025	2.211	0.103	0.348	0.288	4
	*3yBW02*	LG04	HDSC00222_3802	117.623	2.972	0.136	0.375	0.305	1
	*3yBW03*	LG06	HDSC01200_100472	64.852	2.211	0.103	0.392	0.315	6
	*3yBW04*	LG08	HDSC00069_17588	28.499	2.194	0.102	0.340	0.282	2
	*3yBW05*	LG11	HDSC01529_541097	34.812	2.242	0.104	0.499	0.375	2
	*3yBW06*	LG11	HDSC01433_80411	35.054	2.326	0.108	0.500	0.375	2
	*3yBW07*	LG12	HDSC03732_121812	70.449	3.415	0.154	0.287	0.246	4
	*3yBW08*	LG12	HDSC02262_492017	71.907	3.204	0.145	0.379	0.307	2
	*3yBW09*	LG12	HDSC01679_243975	74.531	3.208	0.145	0.291	0.248	5
	*3yBW10*	LG12	HDSC00532_295313	77.767	3.283	0.149	0.297	0.253	1
	*3yBW11*	LG13	HDSC00999_240075	45.318	2.027	0.095	0.392	0.315	2
	*3yBW12*	LG13	HDSC00040_875405	66.641	2.177	0.101	0.498	0.374	5
	*3yBW13*	LG13	HDSC00123_226127	66.679	2.255	0.105	0.258	0.225	5
	*3yBW14*	LG13	HDSC00705_579444	67.158	2.503	0.115	0.356	0.293	1
	*3yBW15*	LG13	HDSC00023_1297306	67.53	2.096	0.098	0.364	0.298	1
	*3yBW16*	LG13	HDSC00539_43988	67.537	2.093	0.097	0.348	0.288	3
	*3yBW17*	LG13	HDSC00706_279669	75.196	2.000	0.093	0.496	0.373	7

(*^1^) LOD: logarithm of the odds; (*^2^) PVE: percent variance explained; (*^3^) He: heterozygosity; (*^4^) PIC: polymorphism information content; (*^5^) No. of genes found in the QTL flanking region; (*^6^) The first-year data were adapted from Kho et al., 2021 [7].

**Table 5 ijms-24-13388-t005:** Quantitative trait loci (QTLs) for shell length (SL) in a full-sib family of Pacific abalone at different ages.

Age	QTL	Linkage Group	Nearest Marker	Position (cM)	LOD *^1^	PVE *^2^	He *^3^	PIC *^4^	No. of Genes *^5^
*First Year *^6^*									
	*1ySL01*	LG10	HDSC01365_168769	0.000	13.490	0.794	0.375	0.305	3
	*1ySL02*	LG10	HDSC01726_79417	8.104	16.513	0.823	0.386	0.311	1
*Second Year*									
	*2ySL01*	LG02	HDSC00677_150547	63.714	3.470	0.156	0.385	0.311	1
	*2ySL02*	LG02	HDSC00607_271588	63.759	3.473	0.156	0.331	0.276	5
	*2ySL03*	LG06	HDSC00436_228138	50.751	3.003	0.136	0.408	0.325	2
	*2ySL04*	LG07	HDSC00922_267573	19.045	3.436	0.155	0.404	0.322	1
	*2ySL05*	LG07	HDSC00134_290236	19.118	3.380	0.152	0.395	0.317	2
*Third Year*									
	*3ySL01*	LG04	HDSC00222_3802	117.623	2.991	0.136	0.375	0.305	1
	*3ySL02*	LG06	HDSC01300_285668	74.819	2.575	0.119	0.405	0.323	5
	*3ySL03*	LG06	HDSC00412_566595	79.907	2.887	0.132	0.227	0.201	1
	*3ySL04*	LG08	HDSC00069_17588	28.499	2.753	0.126	0.340	0.282	2
	*3ySL05*	LG10	HDSC00130_23920	17.46	3.188	0.145	0.379	0.307	2
	*3ySL06*	LG10	HDSC02582_220949	17.985	2.776	0.127	0.405	0.323	4
	*3ySL07*	LG11	HDSC01448_29264	34.477	2.533	0.117	0.496	0.373	2
	*3ySL08*	LG11	HDSC01529_541097	34.812	2.865	0.131	0.499	0.375	2
	*3ySL09*	LG11	HDSC01433_80411	35.054	2.808	0.129	0.500	0.375	2
	*3ySL10*	LG12	HDSC03732_121812	70.449	3.690	0.165	0.287	0.246	4
	*3ySL11*	LG12	HDSC02262_492017	71.907	3.731	0.167	0.379	0.307	2
	*3ySL12*	LG12	HDSC01679_243975	74.531	3.757	0.168	0.291	0.248	5
	*3ySL13*	LG12	HDSC00532_295313	77.767	3.851	0.172	0.297	0.253	1

(*^1^) LOD: logarithm of the odds; (*^2^) PVE: percent variance explained; (*^3^) He: heterozygosity; (*^4^) PIC: polymorphism information content; (*^5^) No. of genes found in the QTL flanking region; (*^6^) The first-year data were adapted from Kho et al., 2021 [7].

**Table 6 ijms-24-13388-t006:** Quantitative trait loci (QTL) for shell width (SW) at different ages in a full-sib family of Pacific abalone.

Age	QTL	Linkage Group	Nearest Marker	Position (cM)	LOD *^1^	PVE *^2^	He *^3^	PIC *^4^	No. of Genes *^5^
*First Year *^6^*									
	1ySW01	LG10	HDSC01365_168769	0.000	30.886	0.735	0.375	0.305	3
	1ySW02	LG10	HDSC01726_79417	8.104	33.883	0.773	0.386	0.311	1
*Second Year*									
	2ySW01	LG02	HDSC00677_150547	63.714	4.741	0.207	0.385	0.311	1
	2ySW02	LG02	HDSC00607_271588	63.759	4.763	0.208	0.331	0.276	5
*Third Year*									
	3ySW01	LG02	HDSC01658_164332	26.222	2.593	0.119	0.356	0.293	1
	3ySW02	LG02	HDSC02578_525042	26.367	2.561	0.118	0.379	0.307	3
	3ySW03	LG04	HDSC00222_3802	117.623	2.928	0.134	0.375	0.305	1
	3ySW04	LG06	HDSC00412_566595	79.907	2.583	0.119	0.227	0.201	1
	3ySW05	LG10	HDSC00130_23920	17.460	2.906	0.133	0.379	0.307	2
	3ySW06	LG10	HDSC02582_220949	17.985	2.544	0.117	0.405	0.323	4
	3ySW07	LG11	HDSC01448_29264	34.477	2.843	0.130	0.496	0.373	2
	3ySW08	LG11	HDSC01529_541097	34.812	3.160	0.143	0.499	0.375	2
	3ySW09	LG11	HDSC01433_80411	35.054	3.098	0.141	0.500	0.375	2
	3ySW10	LG12	HDSC03732_121812	70.449	3.800	0.170	0.287	0.246	4
	3ySW11	LG12	HDSC02262_492017	71.907	3.817	0.171	0.379	0.307	2
	3ySW12	LG12	HDSC01679_243975	74.531	3.769	0.169	0.291	0.248	5
	3ySW13	LG12	HDSC00532_295313	77.767	3.764	0.168	0.297	0.253	1

(*^1^) LOD: logarithm of the odds; (*^2^) PVE: percent variance explained; (*^3^) He: heterozygosity; (*^4^) PIC: polymorphism information content; (*^5^) No. of genes found in the QTL flanking region; (*^6^) Te first-year data were adapted from Kho et al., 2021 [7].

## Data Availability

The data presented in this study are available in the article and Appendix A. The raw sequence data are available on request from the corresponding author.

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
