# Peer review of "Age-Dependent Growth-Related QTL Variations in Pacific Abalone, Haliotis discus hannai"

_ijms, 2023, doi:10.3390/ijms241713388_

Round 1

Reviewer 1 Report

Review of MS “Age-dependent Growth-related QTL Variations in Pacific Abalone, Haliotis discus hannai” submitted for publication to IJMS by Kho et al.

 The authors identified QTLs and linked SNP markers at different ages that are associated with growth-related traits in Pacific abalone. They found that none of the QTLs detected at any one age were expressed at all three age groups, which means all detected QTLs are age specific. The result was a new discovery, but also brings some challenges to the application guidance of abalone aquaculture.

I provide a detailed list of Minor Revision/suggestions, and I would like to highlight the following ones:

1. Do you have any other abalone families? Have some SNPs corresponding to these QTLs been selected for validation in other families?

 2. In this study, abalone was raised in tanks for the first year, while in the sea cage for the second and third years. Different feeding environments will have different effects on the growth of abalone. Did you have identified QTLs expressed at two and three age groups?

 3. Did you think that none of the QTLs are expressed at all three age groups are caused by the different environment which affect growth, thereby affecting phenotypic data?

 4. Do you think the results of this article can provide guidance for the abalone aquaculture? How?

 5. Table S1 was mistakenly written as Table S11 in Supplementary.

 6.  Korean appears in Tables 2-4, please modify to English.

Author Response

The authors identified QTLs and linked SNP markers at different ages that are associated with growth-related traits in Pacific abalone. They found that none of the QTLs detected at any one age were expressed at all three age groups, which means all detected QTLs are age specific. The result was a new discovery, but also brings some challenges to the application guidance of abalone aquaculture.

I provide a detailed list of Minor Revision/suggestions, and I would like to highlight the following ones:

  1. Do you have any other abalone families? Have some SNPs corresponding to these QTLs been selected for validation in other families?

Reply: We have other abalone families. Now, we are rearing several families, we have a plan to observe whether there are variations in QTL expressions among different families or not. Furthermore, we have a plan to validate SNPs corresponding to the age-specific QTLs obtained in the present study as further work at different years and prepare a new manuscript.

  1. In this study, abalone was raised in tanks for the first year, while in the sea cage for the second and third years. Different feeding environments will have different effects on the growth of abalone. Did you have identified QTLs expressed at two and three age groups?

Reply: Raising abalone in tank (1st year) and sea-cage (2nd year onward) is a common practice in abalone aquaculture system. We follow the general pattern of abalone aquaculture to find accurate results. In plants and fishes, it is reported that QTL could be changed with the change of environment and age. We already mentioned these issues in the discussion section. Although the aquaculture environment in the 2nd and 3rd year was same, QTL was also changed, and there was no identical QTL among these two years.

  1. Did you think that none of the QTLs expressed at all three age groups are caused by the different environment which affect growth, thereby affecting phenotypic data?

Reply: Both environmental and genetical control of growth across ages can affect growth and other phenotypic traits. We have already discussed these issues in the discussion section.

  1. Do you think the results of this article can provide guidance for the abalone aquaculture? How?

Reply: Our group is working on marker-assisted selection breeding on abalone for higher growth. We think the result of this manuscript will provide valuable information on age-specific genetic markers that helps to produce good quality high yielding abalone seed using MAS, which ultimately helps the abalone aquaculture. Further, we think the problem of variable growth of abalone at different ages could be resolved by using age-specific genetic markers in MAS.

  1. Table S1 was mistakenly written as Table S11 in Supplementary.

Reply: Supplementary Table number have been corrected accordingly as Table S1.

  1. Korean appears in Tables 2-4, please modify to English.

Reply: Korean words appeared in supplementary Tables 2-4 have been changed to English.

Reviewer 2 Report

In this manuscript, Kho et al report QTL mapping in commercially relevant size traits of the mollusk Pacific Abalone. The authors sequenced 96 F1 progeny and mapped QTL affecting their size over three years. They found that different QTL affected size traits at different life stages. This is potentially important since marker selected breeding could be used for breeding larger individuals at the most commercially-approariate age.

Major comments:

1) The authors previously published a QTL analysis of year 1 in the same species with the same F1 design. (1) 

While the authors do point out the previous reference, it is not made clear if the study population in year 1 is identical. If it is, then several of the figures of the paper are duplicated from the previous work without indicating that they have already been published. While it makes sense for this comparative analysis to reproduce the results from the previous study, this should be mentioned clearly anywhere that a previously published result of data is used. In any revisions, the authors should clarify which part of their work is new and which part uses previous samples / data in both the methods and the result sections.

2) It is not clear to me why the authors have three genetic maps, given that the QTL analysis appear to have been done in the same individuals over three years. Does the genetic composition of the Pacific Abalone change over the course of its life? If not, why don't the authors combine their sequencing data from the three years to carry out one consistent analysis?

3) How consistent were the three linkage maps? Are they numbered the same allowing direct comparison of the mapping results? The plots in Figure 3 don't appear aligned in terms of linkage group length. 

Minor comment:

4) Given that the authors are able to recover 18 linkage groups, which is the number of chromosomes in the Pacific Abalone (2), could the authors use their linkage map to improve the draft assembly to a chromosome level assembly? Such an analysis could improve the manuscript greatly as well as solve the other methodological issue related to having separate linkage maps.

1. Kho, K. H. et al. Construction of a Genetic Linkage Map Based on SNP Markers, QTL Mapping and Detection of Candidate Genes of Growth-Related Traits in Pacific Abalone Using Genotyping-by-Sequencing. Frontiers in Marine Science 8, (2021).

2. Wang H, Luo X, You W, Dong Y, Ke C. Cytogenetic analysis and chromosomal characteristics of the polymorphic 18S rDNA of Haliotis discus hannai from Fujian, China. PLoS One. 2015 Feb 20;10(2):e0113816. doi: 10.1371/journal.pone.0113816. PMID: 25699679; PMCID: PMC4336138.

The English could use some copy-editing, although the author's intention was clear throughout.

Author Response

In this manuscript, Kho et al report QTL mapping in commercially relevant size traits of the mollusk Pacific Abalone. The authors sequenced 96 F1 progeny and mapped QTL affecting their size over three years. They found that different QTL affected size traits at different life stages. This is potentially important since marker selected breeding could be used for breeding larger individuals at the most commercially-appropriate age.

Major comments:

1) The authors previously published a QTL analysis of year 1 in the same species with the same F1 design (1) . While the authors do point out the previous reference, it is not made clear if the study population in year 1 is identical. If it is, then several of the figures of the paper are duplicated from the previous work without indicating that they have already been published. While it makes sense for this comparative analysis to reproduce the results from the previous study, this should be mentioned clearly anywhere that a previously published result of data is used. In any revisions, the authors should clarify which part of their work is new and which part uses previous samples / data in both the methods and the result sections.

Reply: Thank you for your observation and suggestions. In the materials and method section, it is already mentioned that we have used few data from 1st year analysis in this manuscript which is already published in Kho et al., 2021[1]. 1st year data was used to compare the data over three years. [Page 14, lines 409-411]. According to your suggestion, we have mentioned and point out the reference in each figure and tables in the revised manuscript where previously published data was used.

[1] Kho, K.H.; Sukhan, Z.P.; Hossen, S.; Cho, Y.; Kim, S.C.; Sharker, M.R.; Jung, H.J.; Nou, I.S. Construction of a ge-netic linkage map based on SNP markers, QTL mapping and detection of candidate genes of growth-related traits in Pacific abalone using genotyping-by-sequencing. Front. Mar. Sci. 2021, 8, 713783. https://doi.org/10.3389/fmars.2021.713783

2) It is not clear to me why the authors have three genetic maps, given that the QTL analysis appear to have been done in the same individuals over three years. Does the genetic composition of the Pacific Abalone change over the course of its life? If not, why don't the authors combine their sequencing data from the three years to carry out one consistent analysis?

Reply: Thank you for your suggestion. The goal of this manuscript was to identify the variation of QTLs over the ages. In abalone aquaculture, variable growth of abalone individuals is observed consistently across the ages. In the present experiment, it was also observed that an abalone individual grows slower in 1st year, however in the 2nd and 3rd year, it gain highest growth and vice versa. This could be due to genetic control of growth and due to environmental effects.

Variable growth and heat stress mortality of abalone are the common problems in abalone aquaculture. Due to mortality of abalone, we could not be able to use all same abalone individuals for analysis each year. Each year few abalone were died. In the 2nd and 3rd year anlysis, died abalone individuals was replace by new individual from F1 stock for analysis. However, abalone used in analysis for each year were selected from the same F1 progeny. As we were not used all same abalone individuals across three years that’s why we generated three separate linkage maps. However, most of the manuscripts that observed QTL variations across ages and generate one linkage map have been reported on animals and plants having short life cycle or analyze for short period of time [2,3].

[2] Podisi, B.K.; Knott, S.A.; Burt, D.W.; Hocking, P.M. Comparative analysis of quantitative trait loci for body weight, growth rate and growth curve parameters from 3 to 72 weeks of age in female chickens of a broiler-layer cross. BMC Genet. 2013, 14, 22. https://doi.org/10.1186/1471-2156-14-22

[3] Neuschl, C.; Brockmann, G.A.; Knott, S.A. Multiple-trait QTL mapping for body and organ weights in a cross between NMRI8 and DBA/2 mice. Genet. Res. 2007, 89, 47–59. https://doi.org/10.1017/S001667230700852X

3) How consistent were the three linkage maps? Are they numbered the same allowing direct comparison of the mapping results? The plots in Figure 3 don’t appear aligned in terms of linkage group length.

Reply: The same reference genome was used to generate the linkage maps for each year, and the order of 18 linkage groups for each year was prepared. So that, the same scaffold belongs to the same LG for each year, and it was based on the LG of the first year. Further, since the front part of the middle bar ('_') in the marker name is scaffolds of the draft assembly (Example: HDSC23070_15185 to HDSC23070), LG groups with the same scaffolds were processed in the same order. So that, the consistency of the three linkage maps itself is certain/secured. Figure 3 describes LOD score that identify QTLs based on threshold value. Threshold values are marked by blue lines in each figure over 18 LGs.

Minor comment:

4) Given that the authors are able to recover 18 linkage groups, which is the number of chromosomes in the Pacific Abalone (2), could the authors use their linkage map to improve the draft assembly to a chromosome level assembly? Such an analysis could improve the manuscript greatly as well as solve the other methodological issues related to having separate linkage maps.

Reply: Writing the draft assembly at the chromosome level is another big project itself. The present data generated in this study gives very little information to generate draft assembly at chromosome level. To do this, we need to generate more additional data (eg. Omni-C, Long read, etc.), which is not possible at this stage, and it involves huge financial costs.
